# Critical impacts of interfacial water on C–H activation in photocatalytic methane conversion

Hiromasa Sato [1,2], Atsushi Ishikawa [3], Hikaru Saito [1], Taisuke Higashi[1], Kotaro Takeyasu [4] & Toshiki Sugimoto [1,2,5 ✉]

On-site and on-demand photocatalytic methane conversion under ambient conditions is one of the urgent global challenges for the sustainable use of ubiquitous methane resources. However, the lack of microscopic knowledge on its reaction mechanism prevents the development of engineering strategies for methane photocatalysis. Combining real-time mass spectrometry and *operando* infrared absorption spectroscopy with ab initio molecular dynamics simulations, here we report key molecular-level insights into photocatalytic green utilization of methane. Activation of the robust C–H bond of methane is hardly induced by the direct interaction with photogenerated holes trapped at the surface of photocatalyst; instead, the C–H activation is significantly promoted by the photoactivated interfacial water species. The interfacial water hydrates and properly stabilizes hydrocarbon radical intermediates, thereby suppressing their overstabilization. Owing to these water-assisted effects, the photocatalytic conversion rates of methane under wet conditions are dramatically improved by typically more than 30 times at ambient temperatures (~300 K) and pressures (~1 atm) in comparison to those under dry conditions. This study sheds new light on the role of interfacial water and provides a firm basis for design strategies for non-thermal heterogeneous catalysis of methane under ambient conditions.

[1] Department of Materials Molecular Science, Institute for Molecular Science, Okazaki, Aichi 444-8585, Japan. [2] The Graduate University for Advanced Studies, SOKENDAI, Hayama, Kanagawa 240-0193, Japan. [3] Center for Green Research on Energy and Environmental Materials (GREEN), National Institute for Materials Science (NIMS), Tsukuba, Ibaraki 305-0044, Japan. [4] Faculty of Pure and Applied Sciences, Tsukuba Research Centre for Energy and Materials Science, and R&D Center for Zero CO2 Emission with Functional Materials, University of Tsukuba, Tsukuba, Ibaraki 305-8573, Japan. [5] Precursory Research for Embryonic Science and Technology (PRESTO), Japan Science and Technology Agency (JST), Chiyoda, Tokyo 102-0076, Japan. ✉email: toshiki-sugimoto@ims.ac.jp

Methane, the main component of natural gas and a ubiquitous natural carbon resource, has the most robust C–H bonds (bond dissociation energy: 439 kJ/mol) and the highest activation barrier among the hydrocarbon species[1]. Therefore, conversion of the most unreactive hydrocarbon of methane under mild reaction condition is challenging, remaining one of the globally important agendas for the development of a green society[1]. Industrially, methane is utilized in the catalytic steam reforming reaction ($CH_4 + H_2O \rightarrow CO + 3H_2$) at high temperatures and pressures (700–1100 °C and 20–40 atm) to produce synthesis gas, a feedstock for the production of various chemical products[1–3]. To overcome this energy-intensive process[1,4,5] and realize on-site and on-demand methane conversion technologies for sustainable methane utilization[3], it is indispensable to develop an effective methane activation method at ambient temperatures and pressures.

Photocatalysis is a promising technology in which redox reactions are promoted by light; endergonic reactions such as water splitting[6–8], $CO_2$ reduction[9], and organic synthesis with water[10] can be photocatalyzed at ambient conditions beyond thermodynamic limitations[11]. Recently, the application of photocatalytic technology to the conversion of methane with robust C–H bonds has been reported[12,13]. In principle, the oxidation and reduction reactions are induced on the surface of photocatalysts by photogenerated holes and electrons, respectively. However, despite intensive research in the past, the microscopic mechanism of non-thermal methane activation is not yet clear[6–13].

The lack of an accurate and comprehensive understanding of the microscopic mechanism obscures the design strategies of optimal reaction system for the photocatalytic conversion of methane. Because methane is a fully reduced molecule, its activation is driven solely by the oxidation reaction induced by the photogenerated holes. Various observation methods, such as electron spin resonance[14–18] and fluorescence spectroscopy[18–21], have shown that photogenerated holes can be present on the surfaces of photocatalysts in various forms, such as holes trapped at surface lattice oxygen ($O_{lat}$) sites and holes trapped as surface hydroxyl radicals or ad-atom oxygen radicals derived from adsorbed water species[22–28]. However, definitive identification of the hole-derived reactive species in the non-thermal methane conversion has been difficult with these traditional ex-situ measurement techniques owing to the huge gap between the measurement and actual working conditions for methane photocatalysis. Measurements are generally conducted at liquid-nitrogen temperature[16–18] or in the presence of additive molecules, such as spin trap scavengers[14,15] or radical marker molecules[18–21]. Because these differences in environmental conditions inevitably disturb the inherent reaction system of methane, in-situ/*operando* identification of reactive species and intermediates with non-invasive detection techniques are of crucial importance to elucidate the mechanism of non-thermal C–H activation of methane and to develop engineering strategies for effective interfacial chemical systems at the molecular level.

Here, we combined real-time mass spectrometry with *operando* infrared (IR) absorption spectroscopy and ab initio molecular dynamics (AIMD) simulations to provide microscopic insights into non-thermal photocatalytic conversion of methane. We employed three metal oxides as representative $d^{10}$ (Pt/$Ga_2O_3$) and $d^0$ (Pt/$NaTaO_3$ and Pt/$TiO_2$) photocatalysts[8]. Systematic experiments on these photocatalysts under controlled pressures of methane gas and water vapor clearly showed that interfacial water species play a key role in the photocatalytic methane conversion under ambient conditions. The interfacial water is preferentially oxidized by photogenerated holes trapped at photocatalyst surfaces, and the preactivated water species effectively catalyze the C–H bond breaking of methane. Moreover, the interfacial hydrogen-bond network prevents the overstabilization of the intermediates and contributes to increasing the photocatalytic reactivity of methane at ambient temperatures and pressures.

## Results and discussion

**Impact of interfacial water on the photocatalytic conversion of methane.** We employed $Ga_2O_3$, $NaTaO_3$, and $TiO_2$ as representative $d^{10}$ ($Ga_2O_3$) and $d^0$ ($NaTaO_3$ and $TiO_2$) photocatalysts whose conduction bands are composed predominantly of $sp$ and $d$ orbitals, respectively[8]. These photocatalyst samples are known to have stable activities and significant robustness without catalyst deactivation, e.g., photo-corrosion[17,29,30]. In addition, $TiO_2$ is a well-known model material in photocatalysis because the $TiO_2$ photocatalyst has been studied for over half a century[23,31] since the discovery of Honda-Fujishima effect[6].

The reaction activity was evaluated under dry ($P_{H_2O} = 0$ kPa) and wet conditions ($P_{H_2O} = 2$ kPa) at several methane pressures $P_{CH_4}$. Under the reaction conditions with ultraviolet (UV) light irradiation, the sample temperature rises only by approximately 20 K (from ~295 to ~318 K). As discussed in detail in Supplementary Note 1, one layer of adsorbed water molecules covers the sample under the wet condition ($P_{H_2O} = 2$ kPa) at 318 K (corresponding to ~20% relative humidity)[22].

The methane conversion rates under the dry and wet conditions at $P_{CH_4} = 70$ kPa for the Pt/$Ga_2O_3$ sample are compared in Fig. 1a. Under the dry condition, only ethane and hydrogen were detected, indicating the occurrence of photocatalytic non-oxidative methane coupling ($2CH_4 \rightarrow C_2H_6 + H_2$)[32–34]. In this case, the conversion rate was extremely low (~0.2 μmol/h). By contrast, in the presence of adsorbed water layer under the wet condition, the methane conversion rates drastically increased (Fig. 1a). Not only the production of carbon dioxide and hydrogen ($CH_4 + 2H_2O \rightarrow CO_2 + 4H_2$) but also the production of carbon monoxide ($CH_4 + H_2O \rightarrow CO + 3H_2$) and ethane ($2CH_4 \rightarrow C_2H_6 + H_2$) was confirmed as methane derived species (Fig. 1a and Fig. S2–1 in Supplementary Note 2). The total $CH_4$ conversion rate to carbon-containing products ($CO_2$, CO, and $C_2H_6$) was 11.6 ± 0.3 μmol/h, which is over 30 times higher than the value obtained in the absence of water. The hydrogen formation rate also increased under the wet condition (Fig. 1b), which is consistent with the enhanced methane conversion. Nearly identical water-assisted activity enhancements were also observed for the Pt/$NaTaO_3$ (Fig. 1c, d) and Pt/$TiO_2$ (Fig. 1e, f) photocatalysts, indicating that interaction of methane with interfacial water plays a key role in the non-thermal C–H activation and conversion of methane, which would be an independent feature of the $d^{10}$ and $d^0$ photocatalyst materials.

To verify the contribution of interfacial adsorbed water on methane conversion, reaction experiments were conducted for all photocatalysts using isotopically labeled water ($H_2^{18}O$). Note that the methane conversion rate was independent of water isotopes. As shown in Fig. 1g–i, the ratio of $^{18}O$ species to $^{16}O$ species indicates that the contribution of interfacial water ($^{18}O$ species) to $CO_2$ formation is substantially dominant to that of $O_{lat}$ ($^{16}O$ species) regardless of the photocatalyst sample used. This indicates that the photocatalytic oxidative methane conversion is not mainly induced by photogenerated holes trapped at surface $O_{lat}$ sites; instead, it is dominated by interfacial water species preactivated with photogenerated holes. This observed sample-independent feature is in stark contrast to previous studies on the pure water splitting reaction, for which the involvement of $O_{lat}$ in photocatalysis depends on whether photocatalyst material is $d^{10}$ (Pt/$Ga_2O_3$[35]) or $d^0$ (Pt/$NaTaO_3$[36] and Pt/$TiO_2$[37]).

Notably, ethane production was also dramatically enhanced for all photocatalysts owing to the presence of interfacial water

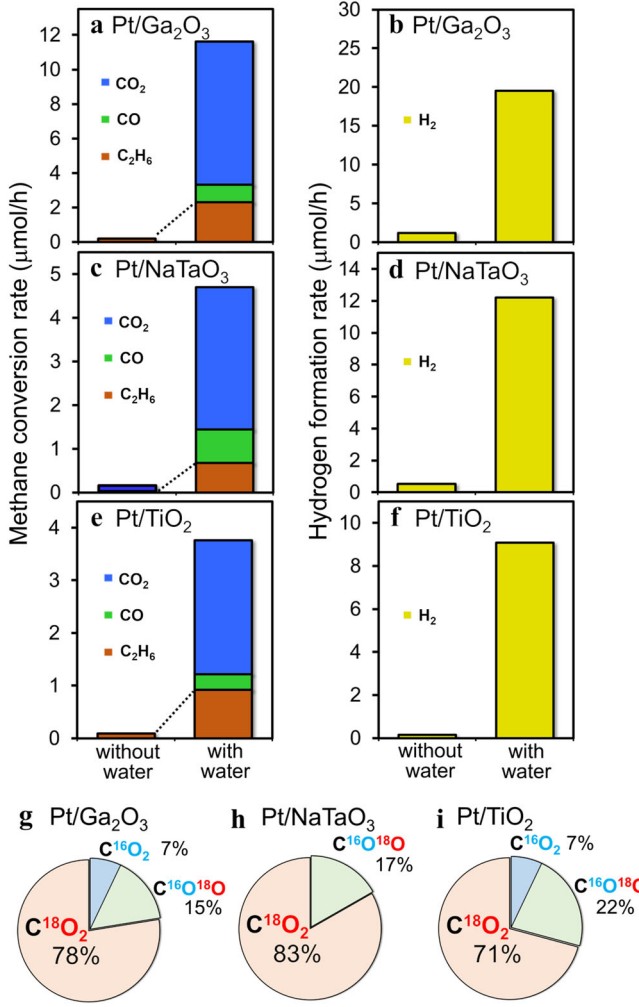

**Fig. 1 Effect of interfacial water on the photocatalytic conversion of methane.** Methane conversion rates and hydrogen formation rates on the **a**, **b** Pt/Ga$_2$O$_3$, **c**, **d** Pt/NaTaO$_3$, and **e**, **f** Pt/TiO$_2$ photocatalysts under ultraviolet irradiation at a methane partial pressure of 70 kPa and water (H$_2^{18}$O) partial pressures of 0 and 2 kPa. Ratio of the produced carbon dioxide species (C$^{16}$O$_2$, C$^{16}$O$^{18}$O, and C$^{18}$O$_2$) on **g** Pt/Ga$_2$O$_3$, **h** Pt/NaTaO$_3$, and (**i**) Pt/TiO$_2$ photocatalysts at a H$_2^{18}$O partial pressure of 2 kPa.

(Fig. 1a, c, and e), although water itself is not involved in the equation of methane coupling reaction ($2CH_4 \rightarrow C_2H_6 + H_2$). Thus, ethane formation under wet conditions must be mechanistically different from the non-oxidative coupling of methane previously reported in the absence of water[32–34]. This implies that the interfacial water species contribute by catalyzing the first oxidative C–H cleavage process ($CH_4 \rightarrow \cdot CH_3$) and promote the subsequent homocoupling ($2 \cdot CH_3 \rightarrow C_2H_6$)[13,17].

**Direct spectroscopic evidence of water-mediated C–H cleavage.** To shed new light on the role of interfacial water in the C–H cleavage process, we conducted *operando* IR spectroscopy with isotope-labeled water (D$_2$O) under the reaction conditions. Almost no isotope effect was observed on the conversion rate and selectivity (Fig. S2-2), indicating that isotope labeling did not affect the photocatalytic process of methane. Figure 2a–c shows the time evolution of the IR spectra in the O–H stretching region under UV irradiation ($P_{CH_4} = 30$ kPa; wet conditions, $P_{D_2O} = 2$ kPa) for Pt/Ga$_2$O$_3$, Pt/NaTaO$_3$, and Pt/TiO$_2$. The O–H stretching peak (3000–3600 cm$^{-1}$) derived from the hydrogen-

bonded adsorbed HDO molecules emerged under UV irradiation (see also Supplementary Note 3 for details). Note that the growth of the O–H peak on Pt/TiO$_2$ photocatalysts was more clearly observed than the other samples due to the quite small particle size (see Fig. S4-1 in Supplementary Note 4) and the large surface area. In good agreement with the linear time evolution of the products (Figs. S2-1 and S2-3), the intensity of the O–H peak increased linearly with irradiation time (Fig. 2d–f). No appreciable spectroscopic change was observed under dark conditions. The increase in the amount of adsorbed HDO thus clearly indicates hydrogen abstraction from methane by OD radicals ($\cdot$OD) at the photocatalyst surface, as follows:

$$CH_{4(gas)} + \cdot OD_{(ad)} \rightarrow \cdot CH_{3(ad)} + HDO_{(ad)}. \quad (1)$$

The $\cdot$OD$_{(ad)}$ is preferentially formed via the oxidation of adsorbed D$_2$O molecule by the surface-trapped holes at O$_{lat}$ site[23,24] as follows:

$$D_2O_{(ad)} + h^+_{(O_{lat})} \rightarrow \cdot OD_{(ad)} + D^+. \quad (2)$$

If the CH$_3$ radical ($\cdot$CH$_3$) in Eq. (1) is the first surface intermediate on the methane photocatalysis, then the peak growth rate of HDO should reflect the methane conversion rate. We conducted additional *operando* IR spectroscopy studies at various methane partial pressures (Fig. 2d–f) to confirm this relationship. As shown in Fig. 2g–i, the HDO peak growth rate sharply increased with $P_{CH_4}$ below 30 kPa and was saturated at higher $P_{CH_4}$. In good agreement with the IR spectroscopy results, the total conversion rate of methane also increased sharply with $P_{CH_4}$ below 30 kPa (~0.3 atm) and was nearly saturated at $P_{CH_4} = 100$ kPa (~1 atm) regardless of the photocatalyst sample used. This correlation between the IR observation and methane conversion indicates that the hydrogen abstraction process on photocatalyst surfaces by photoactivated interfacial water species is the initial key step in methane photocatalysis under wet conditions. This is in contrast to thermal oxidation of methane, in which $\cdot$OH radical is released in gas phase and then methane is activated by the free $\cdot$OH radical ($CH_{4(gas)} + \cdot OH_{(gas)} \rightarrow \cdot CH_{3(gas)} + H_2O_{(gas)}$)[38].

The pronounced increase in the ethane production rate under the wet conditions (Fig. 1) can be rationally explained based on the insights into the initial C–H cleavage process ($CH_4 \rightarrow \cdot CH_3$). Under dry conditions, C–H cleavage is induced via hydrogen abstraction only by direct hole transfer from surface O$_{lat}$. The presence of interfacial water offers another C–H cleavage process via hydrogen abstraction by photoactivated interfacial water species (Eq. (1'); $CH_{4(gas)} + \cdot OH_{(ad)} \rightarrow \cdot CH_{3(ad)} + H_2O_{(ad)}$). Owing to the water-assisted process, the production of reactive $\cdot$CH$_3$ radicals would become more efficient, and the homocoupling reaction ($2\cdot CH_3 \rightarrow C_2H_6$) is accelerated under wet conditions.

Notably, the kinetic isotope effects on methane conversion and total hydrogen production were negligible (Fig. S2-2). This result indicates that water activation (Eq. (2)) does not determine the reaction rate in the methane photocatalytic reactions, which is contrary to water splitting, where water activation has been considered to be a rate-determining step[39]. Since the redox potential of water oxidation ($E^\circ_{\cdot OH/H_2O} = 2.73$ V vs. the standard hydrogen electrode (SHE) at pH7)[40] is higher than that of methane oxidation ($E^\circ_{\cdot CH_3/CH_4} = 2.06$ V vs. SHE at pH7)[41], it is simply assumed from the thermodynamic point of view that the photogenerated holes get more stabilized by the oxidation of methane than water. In contrast to the thermodynamic tendency, however, our experimental results

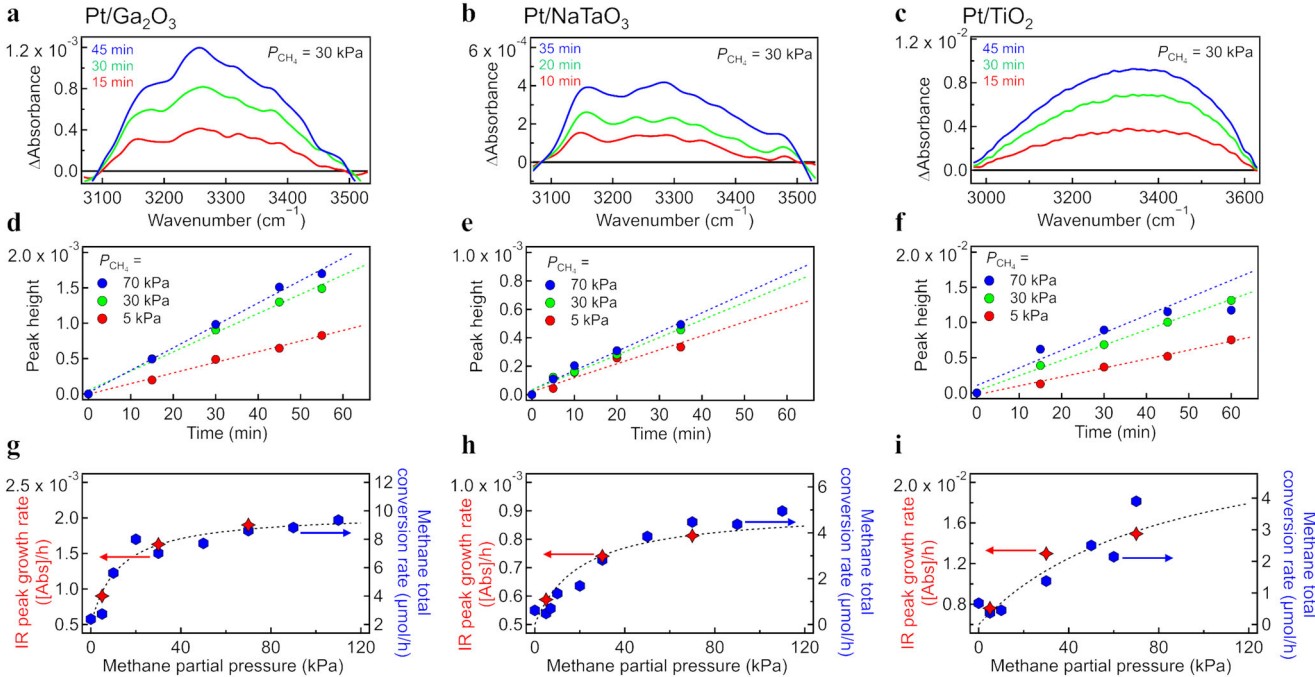

**Fig. 2 *Operando* infrared (IR) spectroscopy of photocatalytic methane activation with isotope-labeled water (D$_2$O).** Time evolution of the IR spectra in the O–H stretching region for the **a** Pt/Ga$_2$O$_3$, **b** Pt/NaTaO$_3$, and **c** Pt/TiO$_2$ photocatalysts under ultraviolet irradiation at a CH$_4$ pressure of 30 kPa under wet conditions (2 kPa of D$_2$O). The *operando* IR measurements were started from the time when the temperature increase leveled off in order to extract the response derived from photocatalytic reaction while excluding the spectral change derived from the sample heating (see Supplementary Note 9 for details). Time zero ($t = 0$ min) was defined as the starting time for the IR measurement. Time evolution of the peak height at 3250 cm$^{-1}$ for the **d** Pt/Ga$_2$O$_3$, **e** Pt/NaTaO$_3$, and **f** Pt/TiO$_2$ photocatalysts at CH$_4$ pressures of 5, 30, 70 kPa. Growth rate of the peak at 3250 cm$^{-1}$ (left axis) and CH$_4$ total conversion rate (right axis) on the **g** Pt/Ga$_2$O$_3$, **h** Pt/NaTaO$_3$, and (**i**) Pt/TiO$_2$ photocatalysts.

(Figs. 1 and 2) indicate that the holes at the oxide surfaces preferentially oxidize water rather than methane, and then the preactivated water species activate the C–H bond of methane. This implies that photocatalytic preferential oxidation of water over methane by the surface holes is induced kinetically rather than thermodynamically.

**AIMD simulations of photocatalytic C–H activation processes.** To provide detailed molecular-level insights into the photocatalytic C–H activation kinetics, AIMD simulations were carried out with density functional theory (DFT) calculations with hybrid fuctional on a β-Ga$_2$O$_3$ surface. Our hybrid DFT calculations can adequately treat neutral and charged (or hole) states that are presumably involved in the pristine reaction. The details of calculation procedure are presented in the "Methods" section.

First, we confirmed in our calculations that a hole exists in the O$_{lat}$ site on the bare Ga$_2$O$_3$ surface (Fig. S5-1 in Supplementary Note 5). Therefore, under dry conditions, ·CH$_3$ is assumed to be generated via the direct interaction of CH$_4$ with the hole trapped at the surface O$_{lat}$ site. The transfers of the proton and the hole (electron) would occur simultaneously via proton-coupled electron transfer (PCET). Then, we examined the potential energy curve (PEC) for the ·CH$_3$ formation and found that the ·CH$_3$ formation is exothermic ($\Delta E = -90.6$ kJ/mol, Figs. S5-3a and S5-3b) and that the generated ·CH$_3$ is subsequently adsorbed on the Ga$_2$O$_3$ surface with a large adsorption energy ($E_{ads} = -171.9$ kJ/mol, Fig. S5-3c). This overstabilization hinders the homocoupling of ·CH$_3$ to form ethane (2·CH$_3 \rightarrow$ C$_2$H$_6$) and results in further hydrogen abstraction from ·CH$_3$ to form other hydrocarbon species and coke[32–34].

This overstabilization is strongly altered by hydration with interfacial water. Under wet conditions, the Ga$_2$O$_3$ surface is covered with several layers of adsorbed water, which is preferentially activated by interacting with the hole center on the O$_{lat}$ (H$_2$O + O$_{lat(h^+)} \rightarrow$ ·OH + H–O$_{lat}$) via PCET (Fig. S5-2). Our calculations showed that the generation of ·OH has an activation barrier of 21.2 kJ/mol and that the reaction is moderately exothermic ($\Delta E = -14.9$ kJ/mol), which indicates that surface holes exist as ·OH. Therefore, under the wet condition, the water-assisted activation pathway of methane emerges: CH$_4$ + ·OH $\rightarrow$ ·CH$_3$ + H$_2$O. Figure 3 shows the PEC and snapshots of the MD trajectory of the reaction between CH$_4$ and ·OH. The C–H activation of methane via hydrogen abstraction by ·OH has an energy barrier of 38.4 kJ/mol, which can be adequately overcome at ambient temperature[42,43]. In this case, the produced ·CH$_3$ is moderately stabilized by the hydration with water ($\Delta E = -50.2$ kJ/mol). The proper stabilization of the reaction intermediates in the order of 50 kJ/mol is crucial for the methane conversion to proceed under ambient reaction conditions (~300 K, ~1 atm), which is discussed in detail in the following section.

Our calculations showed that methane was more stabilized through the activation process ($\Delta E \sim -50$ kJ/mol under the wet condition, Fig. 3a) than water ($\Delta E \sim -15$ kJ/mol, Fig. S5-2). This result agrees well with the difference in redox potential[40,41]. By contrast, the barrier for water activation is ~20 kJ/mol (Fig. S5-2), which is much lower than that for methane activation under the wet condition (~40 kJ/mol, Fig. 3a). Thus, water is kinetically more easily activated than methane, although methane is thermodynamically more oxidizable than water. This kinetic advantage and the relatively high population around the active sites would contribute to the preferential water oxidation on C–H activation process of methane.

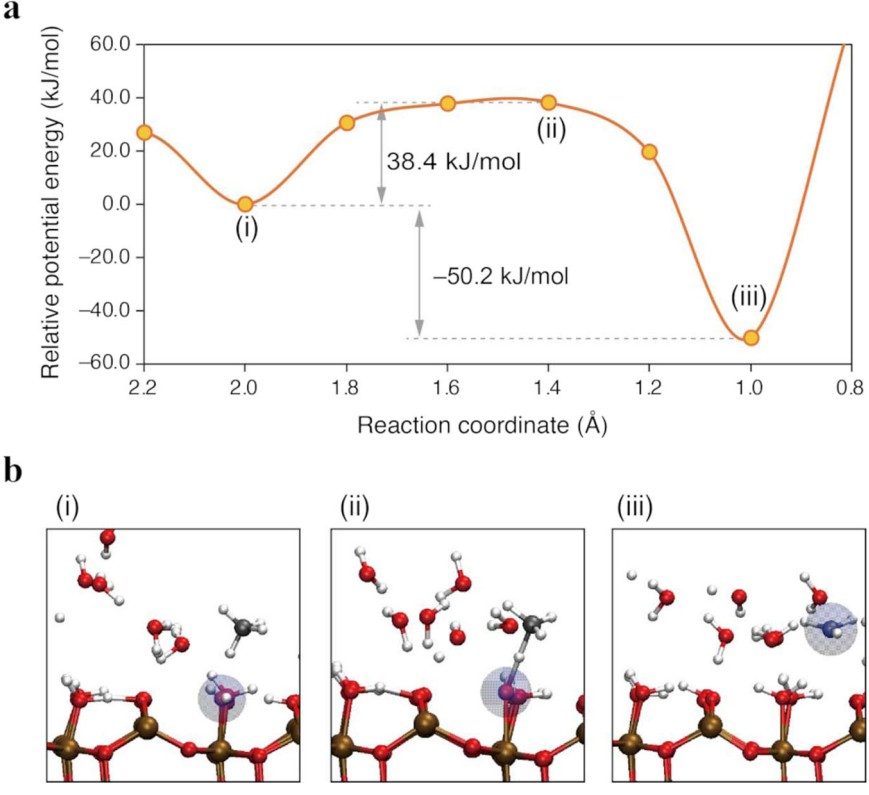

**Fig. 3 Ab initio molecular dynamics simulation results. a** Potential energy curve (PEC) and **b** snapshots corresponding to the dots on PEC (i–iii) for methane activation under wet conditions on a $\beta$-$Ga_2O_3$ surface. The blue-shaded spheres in the snapshots indicate the hole, which is identified from a Mulliken spin population value larger than 0.5 e.

**Unveiling the reaction conditions that maximize photocatalytic activity.** The microscopic properties of the interfacial reaction system discussed above directly affect the macroscopic reaction kinetics and optimal reaction conditions. As shown in Fig. 2g–i, the methane conversion rates under the wet conditions increased with an increase in $P_{CH_4}$ below 1 atm and were saturated at $P_{CH_4} = 1$ atm. As discussed in this section, the moderate stabilization of the •$CH_3$ radical intermediate revealed by AIMD simulations (Fig. 3) has direct consequences on the maximization of photocatalytic performance under ambient pressures (~1 atm).

The methane conversion rates were divided into the formation of each product ($H_2$, $CO_2$, CO, and $C_2H_6$). As shown in Figs. 4 and S2-4, the formation rates increase with $P_{CH_4}$ and are maximized at approximately 1 atm. If the first C–H cleavage process (Eq. (1'')) is the rate-determining step (as considered for the thermocatalytic steam reforming[44,45]), the formation rates would increase linearly with $P_{CH_4}$. However, the rates do not show linear $P_{CH_4}$ dependences (Fig. 4), indicating that the first C–H activation process is not rate-determining on photocatalytic methane conversion. This non-linear behavior is sample-independently observed for $Pt/Ga_2O_3$, $Pt/NaTaO_3$, and $Pt/TiO_2$ photocatalysts, indicating that the reaction mechanism is common in the three samples and that methane partial pressure is a key parameter for the occurrence of non-thermal methane conversion.

As discussed in the previous section, photoactivation of interfacial water is not rate-determining and the methane from the gas phase is cleaved by these photo-activated water species (Eq. (1'); $CH_{4(gas)} + •OH_{(ad)} \rightarrow •CH_{3(ad)} + H_2O_{(ad)}$) with a relatively low activation barrier (Fig. 3a). The fate of the adsorbed methyl radical intermediate (denoted as $X_1$ in Fig. 5a, b, and Fig. S6-1 in Supplementary Note 6) is twofold: it desorbs to the

gas phase as a methane molecule via the backward reaction from Eq. (1') (•$CH_{3(ad)} + H_2O_{(ad)} \rightarrow CH_{4(gas)} + •OH_{(ad)}$) or undergoes multistep surface reactions to form $C_2H_6$, CO, and $CO_2$ through multiple intermediates (Fig. 5a, b). In the case where the reaction of the adsorbed •$CH_3$ is rate-determining, then the formation rate ($R$) of each molecule under steady-state reaction condition is simply described by the following equation: $R \propto [\theta_{CH_3}]^n$ (see Supplementary Notes 6-1 and 6-2), where $\theta_{CH_3}$ is the surface coverage of the •$CH_{3(ad)}$ intermediate and $n$ is the reaction order: $n = 1$ for the CO and $CO_2$ formations and $n = 2$ for the $C_2H_6$ formation.

As discussed in detail in Supplementary Note 6, the $P_{CH_4}$ dependence of $\theta_{CH_3}$ is given by:

$$\theta_{CH_3} = \frac{KP_{CH_4}}{1 + KP_{CH_4}}, \qquad (3)$$

where $K$ is the equilibrium constant for the forward (dissociative adsorption of methane) and reverse (desorption of methane) reactions of Eq. (1'), expressed by: $K \equiv k_{ad}/k_{de} \propto \exp(U/k_BT_s)$. $k_B$ and $T_s$ are the Boltzmann constant and surface temperature, respectively, and $U$ is the stabilization energy of the •$CH_{3(ad)}$ intermediate ($X_1$) relative to that of methane in the presence of interfacial water (Fig. S6-2). The $P_{CH_4}$ profile of $\theta_{CH_3}$ is similar to the Langmuir adsorption isotherm on a logarithmic pressure scale (Fig. 5c). As shown in Fig. 5c, the threshold pressure for $\theta_{CH_3}$ significantly decreases from $10^5$ kPa ($10^3$ atm) to $10^1$ kPa ($10^{-1}$ atm) as $U$ increases from 15 kJ/mol to 40 kJ/mol. The considerable difference between the dependences of $\theta_{CH_3}$ and $\theta_{CH_3}^2$ on $P_{CH_4}$ is shown in Fig. S6-4a on a linear pressure scale: $\theta_{CH_3}$ increases as an upward convex curve and then saturates, whereas $\theta_{CH_3}^2$ increases sigmoidally.

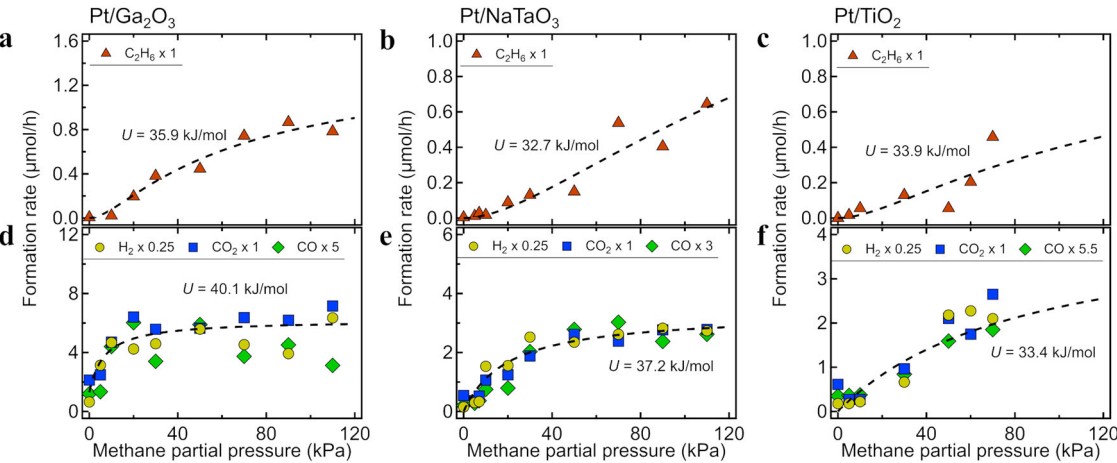

**Fig. 4 $P_{CH_4}$ profiles of photocatalytic methane conversion with water.** Formation rates of $H_2$, $CO_2$, CO, and $C_2H_6$ for the **a**, **d** Pt/Ga$_2$O$_3$, **b**, **e** Pt/NaTaO$_3$, and **c**, **f** Pt/TiO$_2$ photocatalysts under UV irradiation as functions of methane partial pressure at a $H_2O$ partial pressure of 2 kPa. The dashed lines are the curve fitting results following the equation: $[KP_{CH_4}/(1 + KP_{CH_4})]^n$, where $n = 2$ for $C_2H_6$ and $n = 1$ for CO and $CO_2$.

Based on these characteristic $P_{CH_4}$ profiles, the reaction order $n$ can be clearly distinguished, and the stabilization energy $U$ can be evaluated by curve fitting. For ethane formation, the reaction intermediate is assumed to be $^\bullet$CH$_{3(ad)}$, while the reaction order $n$ is 2 for the homocoupling. As shown in Fig. 4a–c, the ethane production rates for the three photocatalyst samples increased sigmoidally with $P_{CH_4}$; these curves are well described by the square of the Langmuir adsorption isotherm (Eq. (3)) as $R_{C_2H_6} \propto \theta_{CH_3}^2 = [KP_{CH_4}/(1 + KP_{CH_4})]^2$, with the $U$ values ~40 kJ/mol (Table 1). These are in good agreement with the stabilization energy obtained from the theoretical calculations (Fig. 3a). Notably, if the $^\bullet$CH$_3$ radical intermediate is subsequently released into the gas phase and the homocoupling reaction proceeds there, the production rate of ethane is given by the apparent first-order reaction with respect to the coverage of surface adsorbed $^\bullet$CH$_3$ intermediate species as follows: $R_{C_2H_6} \propto \theta_{CH_3} = KP_{CH_4}/(1 + KP_{CH_4})$ (see Supplementary Note 6-3 for details). Therefore, the experimentally observed dependence of $R_{C_2H_6}$ on $\theta_{CH_3}^2$ (Fig. 4a–c) indicates that the homocoupling reaction occurs on the surfaces of water-covered photocatalyst samples at a low temperature (~318 K).

For CO and $CO_2$, the observed $P_{CH_4}$ dependences (Fig. 4d–f) are well described by first-order expressions ($n = 1$) as $R_j \propto KP_{CH_4}/(1 + KP_{CH_4})$ ($j$ = CO or $CO_2$) with the $U$ values listed in Table 1. Notably, CO and $CO_2$ are formed through multiple reactions with various intermediates (Figs. 5b and S6-1c). Some of the surface intermediates were detected by *operando* IR spectroscopy in the C–H and C–O stretching regions (Fig. S7-1 in Supplementary Note 7).

Consistent with our theoretical calculations (Fig. 3a), the stabilization energy in the order of 40 kJ/mol is considerably higher than the typical physisorption energy of a methane molecule (~15 kJ/mol, see Table S8-1 in Supplementary Note 8). Figure 5c clearly shows that the threshold methane pressure decreases as the stabilization energy of the adsorbed intermediate increases; the threshold pressure becomes extremely high (~1000 atm) if the photocatalytic methane conversion is initiated by the molecularly physisorbed intermediate with an adsorption energy of ~15 kJ/mol; meanwhile, the value is comparable to or lower than 1 atm if the conversion is initiated by the dissociatively chemisorbed $^\bullet$CH$_3$ intermediates with an adsorption energy of ~40 kJ/mol.

Under the dry conditions without interfacial water, the photocatalytic methane conversion was hardly induced (Fig. 1), possibly because of the overstabilization of the intermediates (~90 kJ/mol, Fig. S5-3a); the highly stabilized and strongly adsorbed C–H activated species would be overoxidized and thus cause deactivation of the photocatalyst. This scenario is strongly supported by previous experimental observations in which the photocatalyst was covered with carbonaceous deposit and exhibited poor performance under dry reaction conditions[32–34]. In comparing these experimental and theoretical results under dry conditions, we can conclude that the interaction of methane with interfacial water plays a key role not only in substantially lowering the barrier for the C–H cleavage but also in preventing the overstabilization of surface intermediate radical species, resulting in the substantial promotion of photocatalytic methane conversion under ambient conditions.

Note that even if the rate-determining steps are the reactions of the intermediates $X_i$ ($i \geq 2$) at the later part of the multiple surface reactions (Fig. S6-1c) instead of the forward reaction of the $^\bullet$CH$_3$ radical ($X_1$), our main conclusion regarding the water-assisted effects is essentially unaffected. As discussed in detail in Supplementary Note 6-2, similar expressions are obtained for the CO and $CO_2$ formation rates vs. $P_{CH_4}$: $R_j \propto KP_{CH_4}/(1 + KP_{CH_4})$ ($j$ = CO or $CO_2$), where the constant $K$ is proportional to $\exp(U/k_B T_s)$ and $U$ is the stabilization energy of the corresponding rate-determining intermediate (Fig. S6-6).

We remark finally that the effect of the interfacial water on photocatalytic methane activation would be a common phenomenon for most of the metal oxide photocatalysts. Since valence bands of metal oxides mainly consist of O2p orbitals, the valence band maximums of metal oxides exist at a similar depth (~3 eV from the SHE). Furthermore, most of the metal oxide surfaces interact with methane molecules quite weakly in comparison to water molecules: thus, methane molecules cannot substantially adsorb on surfaces without UV irradiation. Therefore, the interfacial water molecules have much opportunity to accept photogenerated holes from the surfaces in comparison with gaseous methane molecules, and are kinetically advantageous for the hole-driven oxidation. The water-assisted effects would be universal among the metal oxide photocatalysts with these features.

In summary, we have demonstrated for the three representative d$^0$ and d$^{10}$ oxide photocatalysts (Ga$_2$O$_3$, NaTaO$_3$, and TiO$_2$) with

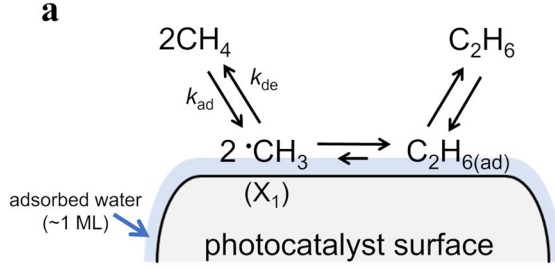

**a**

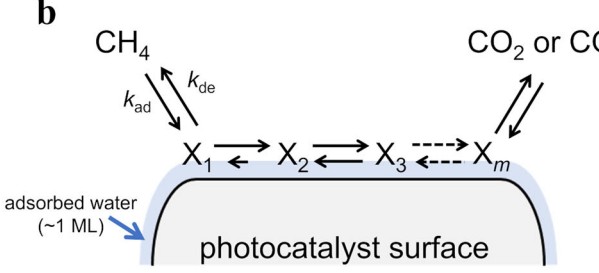

**b**

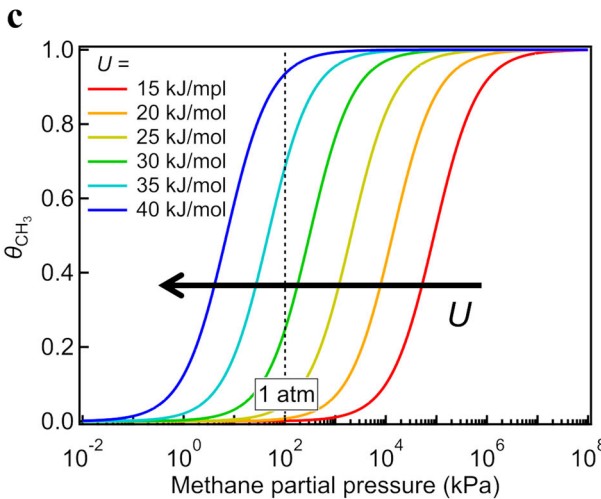

**c**

**Fig. 5 Reaction scheme of the photocatalytic methane conversion.** Surface reaction scheme of the photocatalytic conversion of methane to **a** $C_2H_6$ and **b** $CO_2$ or CO. $X_i$ ($i = 1, 2, 3,..., m$) is a reaction intermediate; $X_1$ is the methyl radical (•$CH_3$). **c** Dependence of the coverage of rate-determining methane intermediates ($\theta_{CH_3}$) on $P_{CH_4}$ with various values of the stabilization energy $U$. The horizontal axis is displayed on a logarithmic scale.

**Table 1 Stabilization energy of the methyl radical intermediate (U) for the photocatalytic $C_2H_6$, CO, and $CO_2$ formation.**

|  | Pt/Ga₂O₃ | Pt/NaTaO₃ | Pt/TiO₂ |
|---|---|---|---|
| $U$ (kJ/mol) for $C_2H_6$ formation | 35.9 | 32.7 | 33.9 |
| $U$ (kJ/mol) for CO and $CO_2$ formation | 40.1 | 37.2 | 33.4 |

the different band-gap energy that the photocatalytic activation of the robust C–H bond of methane is efficiently accelerated by interfacial water species at ambient temperatures and pressures. Combining real-time mass spectrometry, *operando* IR spectroscopy, and AIMD simulations, we have shown that the methane conversion is hardly induced by the direct interaction with the trapped hole at the surface $O_{lat}$ site; instead, activation is

significantly promoted by low barrier hydrogen abstraction from methane by the photoactivated interfacial water species. In the water-mediated processes, the photocatalytic C–H activation is not the rate-determining step, which is in stark contrast to the case of traditional thermocatalytic methane reforming. Moreover, owing to the moderate stabilization of •$CH_3$ in the hydrogen-bond network of water, the overall photocatalytic conversion rates are dramatically improved by typically more than 30 times at ambient temperatures (~300 K) and pressures (~1 atm). As essentially opposed to thermal catalysis, methane photocatalysis no longer requires high-pressure methane gas (>20 atm) in the presence of adsorbed water layer. The water-assisted effects are noticeable also in ethane formation, although water is not explicitly involved in the homocoupling reaction equation ($2CH_4 \rightarrow C_2H_6 + H_2$). These results indicate that the interfacial water kinetically plays crucial roles beyond the traditional thermodynamic concept of redox potential, in which oxidation of water by surface trapped holes is less thermodynamically favored than methane oxidation: $E°_{•OH/H_2O} = 2.73$ V[40] and $E°_{•CH_3/CH_4} = 2.06$ V[41] versus SHE at pH7. Our work not only expands the molecular-level understanding of the non-thermal C–H activation and conversion but also provides a fundamental basis for the rational interface design of photocatalytic systems toward the effective and sustainable utilization of methane under ambient conditions.

## Methods

**Reaction activity measurements.** The photocatalytic activities of the catalysts were evaluated using a batch reactor made of stainless steel (SUS304). The catalysts were irradiated with UV light through a $CaF_2$ window in a chamber filled with water vapor ($H_2^{16}O$, $H_2^{18}O$, or $D_2^{16}O$) and methane ($CH_4$). Methane (>99.99% purity) and water vapor ($H_2^{16}O$ ultrapure water, $H_2^{18}O$ water with 98 at.% $^{18}O$, or $D_2^{16}O$ water with 99.9 at.% D) were pre-degassed in an ultrahigh vacuum gas line via freeze-pump–thaw cycles and introduced in the chamber with a base pressure lower than $1 \times 10^{-3}$ Pa. The partial pressure of the water vapor was fixed at 0 kPa or 2 kPa, whereas the partial pressure of methane was varied in the range of 5 kPa (0.05 atm) to 120 kPa (1.2 atm). The light source was a Xe lamp with a high intensity in the deep UV region (UXM-500SX, Ushio Inc.), and light was led to the catalysts using an optical fiber. The intensity of the light irradiated on the catalysts was approximately 90 mW cm$^{-2}$ at a wavelength of 250 nm. The temperature of the samples was measured using a chromel–alumel thermocouple (type-K). The product gases were measured using a quadrupole mass spectrometer (QMS, PrismaPlus QMG220, Pfeiffer Vacuum). For all photocatalysts, the yield of the reaction products increased almost linearly with the UV irradiation time (Figs. S2-1 and S2-3), implying that the photocatalytic reactions proceeded under nearly steady-state conditions. The reaction activities were evaluated using the slopes of the linear fittings of the product yield curves. A carbon dioxide production in the order of 10 µmol corresponds to a consumption of $CH_4$ in the order of 0.1 kPa in the reaction chamber (230 cm³); therefore, the associated decrease in the inlet $CH_4$ pressure can be neglected. Under 2 kPa of water vapor, less than 4% of the adsorbed water was consumed during the reaction experiments (see Supplementary Note 1).

**Catalyst preparation.** The samples used in this study were 1 wt% Pt-loaded commercial $Ga_2O_3$ powder (Pt/$Ga_2O_3$; $Ga_2O_3$ provided by Kojundo Chemical Laboratory Co., Ltd, Japan), 0.2 wt% Pt-loaded La-doped (1 mol%) $NaTaO_3$ (Pt/$NaTaO_3$), and 1 wt% Pt-loaded commercial anatase $TiO_2$ powder (Pt/$TiO_2$; $TiO_2$: ST-01, Ishihara-Sangyo Ltd, Japan). The SEM and TEM images of these photocatalyst samples are shown in Fig. S4-1. The $NaTaO_3$ sample was prepared by a flux method[43]. Pt was loaded onto the $Ga_2O_3$ and $NaTaO_3$ samples using an impregnation method[29,30]. Each sample was soaked in an aqueous $H_2PtCl_6$ solution (corresponding to 1 wt% or 0.2 wt% Pt). The solution was heated to 368 K until dry. The dried samples were calcined at 673 K for 2 h. Pt was loaded on the $TiO_2$ by a photodeposition method[46] using a Xe lamp (UXL-500SX, Ushio Inc.). $TiO_2$ was dispersed in an aqueous ethanol solution (3 vol%) containing $H_2PtCl_6$ (corresponding to 1 wt%) and exposed to light for 1.5 h with continuous stirring. The solution was heated to 393 K until dry. Typical X-ray diffraction patterns of Pt were confirmed. Methane conversion rates for bare photocatalyst sample without Pt cocatalyst were typically less than a tenth of the methane conversion rates for Pt-loaded samples.

***Operando* diffuse reflectance infrared Fourier transform (DRIFT) spectroscopy.** During the reaction experiments, *operando* DRIFT spectroscopy of the catalyst surfaces was conducted using a homemade diffuse scattering measurement

setup[47] coupled with a Fourier transform infrared spectroscopy system (FT/IR-6600GP01, JASCO Ltd.) with a HgCdTe detector at a resolution of 4 cm$^{-1}$. The background for the DRIFT measurements was obtained with each photocatalyst sample at a stable temperature (~318 K) under the reaction conditions (see also Supplementary Note 9 for details).

**Computational details**. The $Ga_2O_3$ was represented by a model with repeated slabs separated by vacuum regions of ~20 Å. The supercell of the $Ga_2O_3$ surface was constructed by repeating (3×2) the $\beta$-$Ga_2O_3$ unit cell in the lateral directions. Water molecules were distributed in the upper and lower parts of the $Ga_2O_3$ surface. To obtain a single $H_2O$ solvation layer on the $Ga_2O_3$, the following procedure was followed: first, 40 $H_2O$ molecules were included in the system. Preliminary MD calculations were performed, and the $H_2O$ molecules that were not included in the first solvation layer were removed, leaving 34 $H_2O$ molecules on the surface. The unit cell of the $H_2O$–$Ga_2O_3$ surface contained 222 atoms; see Fig. S5-1 for the model. The CP2K code (version 6.1) was used for all calculations. The Kohn–Sham equation was solved in a self-consistent manner. Two levels of the DFT setting were used: the generalized gradient approximation (GGA) level and the hybridized DFT level, including the Hatree–Fock exchange. The Perdew–Burke–Ernzerhof (PBE) functional was used for the GGA[48], and the PBE0 was used for the hybrid DFT calculations[49,50]. Hybridized DFT should be considered more accurate because the exact exchange is admixtured. Because the GGA is known to yield a delocalized electronic state, its accuracy (including that of the electron hole), is questionable[26]. Therefore, the PBE functional was used for the preliminary calculation of the ground state, whereas the PBE0 was used for the productive calculations. The MOLOPT basis set and Goedecker–Teter–Hutter pseudopotential were used to represent the valence and core electrons, respectively[51]. The DZVP basis set was used for the valence electrons of all elements. The convergence threshold of the electronic energy was $1.0 \times 10^{-5}$ eV. The cut-off energy of the plane wave was 400 eV. After geometry optimization, an equilibrium MD run was performed for 10 ps at the GGA level. Then, the additional equilibration run at hybridized DFT was conducted for 3 ps, followed by a production run for 7 ps after these equilibration steps. MD simulations were performed with the NVT ensemble, where the Nose–Hoover thermostat was used to control the temperature to 300 K ± 30 K. In the PEC calculation, the difference of the following two distances, namely, (i) between Ga and the O atom of $H_2O$ ($O_{H_2O}$) and (ii) between the H atom of $H_2O$ and the $O_{lat}$, was taken as the reaction coordinate of •OH formation reaction. For the Ga–$O_{H_2O}$ distance, a harmonic-type restraint ($K(x - \text{target})^2$) was applied with the $K$ parameter of $1.0 \times 10^{-3}$ (a.u.). The distance between the O atom of •OH and the H atom of $CH_4$ was considered as the reaction coordinate in the •$CH_3$ formation reaction in the presence of water. The distance between the H atom of $CH_4$ and $O_{lat}$ was considered as the reaction coordinate in the $CH_3$ formation reaction in the absence of water. The average value of the potential energy over the production run (7 ps) was used for the PEC.

## Data availability

All experimental data in this study are available from the corresponding author upon reasonable request.

## Code availability

Python scripts used for analyzing the AIMD results are available at the one of the authors' GitHub page: https://github.com/atsushi-ishikawa/aimd.

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

## Acknowledgements
This work was supported by JST-PRESTO [No. JPMJPR16S7]; JSPS KAKENHI Grant-in-Aid for Specially Promoted Research [No. 17H06087], Grant-in-Aid for Scientific Research (A) [No. 19H00865], Grant-in-Aid for JSPS Fellows [No. 22J13055]; Joint Research by the National Institutes of Natural Sciences (NINS) [No. 01112104]. This work was also partially supported by Demonstration Project of Innovative Catalyst Technology for Decarbonization through Regional Resource Recycling, the Ministry of the Environment, Government of Japan. The authors acknowledge Dr. Akira Yamamoto, Hisao Yoshida for sample preparation and kind support at an early stage in this project, Dr. Kazuyoshi Kanamori for measuring supporting data of adsorption isotherm of methane under dark conditions, and Dr. Fumiaki Kato, Norihiro Aiga, Kazuya Watanabe, Yoshiyasu Matsumoto, Susumu Saito, and Atsunori Sakurai for fruitful discussion.

## Author contributions
T.S. supervised and conceived the overall investigation. H. Sato, H. Saito, T.H., and K.T. prepared the photocatalysts and performed the characterizations, photocatalytic activity tests, and spectroscopic measurements. A.I. performed the MD simulations. All the authors discussed the results and gave comments on the manuscript. H. Sato, A.I. and T.S. wrote the manuscript.

## Competing interests
The authors declare no competing interests.
