## [Peer Review File · Communications Chemistry]

Reviewers' comments:

Reviewer #1 (Remarks to the Author):

In the manuscript, the authors demonstrate the critical role of interfacial water species in photocatalytic methane conversion in Pt/Ga₂O₃, Pt/NaTaO₃ and Pt/TiO₂ materials through a series of characterization techniques and theoretical calculations. Due to these water-assisted effects, the formation of moderately stable •CH₃ radicals in the interface enhances the photocatalytic conversion by more than ~30 times under the conditions of ambient temperature and pressure. However, the quality of the manuscript remains to be improved before its acceptance for publication. Some major comments have been listed for the authors' reference to further improve the quality of this work. Here are some points should be carefully addressed:

1. In this paper, during the photocatalytic methane conversion reaction, not only the environment of the material is different (dry and wet), but the temperature of its surface is also different (from ~295 K to ~318 K). Does temperature have an effect on catalytic activity? How did the authors rule out thermal effects?
2. Due to these water-assisted effects, the conversion efficiency of methane is improved. At the same time, the three materials do not have high selectivity. How does the author think about this?
3. The authors should state in the article the reasons for choosing Pt/Ga₂O₃, Pt/NaTaO₃ and Pt/TiO₂ for research.
4. This sentence mention "The experimental results of preferential water oxidation on the C-H activation process of methane thus imply the crucial role of interfacial water that is unexplainable under conventional simple assumption based on thermodynamic equilibrium." But the oxidation potential of water is higher than that of methane in thermodynamics. Please confirm this view.
5. The authors only studied the role of interfacial water in three materials Pt/Ga₂O₃, Pt/NaTaO₃ and Pt/TiO₂. Then the interfacial water only has an effect on the three materials or is it universal.

Reviewer #2 (Remarks to the Author):

In the manuscript entitled "Critical impacts of interfacial water on the photocatalytic C-H conversion of methane, the authors use IR spectroscopy combined with mass spectrometry supported by simulations to address the role of interfacial water on the conversion of methane over Pt supported on three different support materials.

The manuscript is well-written and authors present a rigorous piece of work and get the maximum out of particularly the IR results. The explain the data in great detail and I cannot discredit these results in any way.

What I really mis in the manuscript is structural information. No structural information is presented at all (XRD, SEM, TEM). Such data would make the story easier to follow and could justify the assumptions in the manuscript. At the very least, include a sketch in the beginning of the manuscript to allow the reader to understand the structural model of the catalytic systems in question.

Recommendation: publish after minor revision.

Reviewer #3 (Remarks to the Author):

The authors combined real-time mass spectrometry with operando infrared (IR) absorption spectroscopy and ab initio molecular dynamics (AIMD) simulations to provide microscopic insights into non-thermal photocatalytic conversion of methane. Systematic experiments on representative photocatalysts (d10 photocatalyst: Pt/Ga₂O₃, d0 photocatalysts: Pt/NaTaO₃ and Pt/TiO₂) under controlled pressures of methane gas and water vapor clearly showed that interfacial water species play a key role in the photocatalytic methane conversion under ambient conditions, but there are some issues needed to be further emphasized before it is suitable for the publication. A major revision is needed, the details as follows:

1. The authors carried out systematic experiments on representative photocatalysts (d10 photocatalyst: Pt/Ga₂O₃, d0 photocatalysts: Pt/NaTaO₃ and Pt/TiO₂), the date of photocatalyst: Pt/TiO₂ should be provided in figure 2.
2. The author should explain the "15 min line" lacking in Fig. 2(c) and the difference between PCH₄=6 in Fig. 2(b) and PCH₄=5 in Fig. 2(d) in this article.
3. The authors think the kinetic isotope effects on methane conversion and total hydrogen production were negligible, the authors should explain why the use of water with and without isotopes in Figure S2-2(b) has a greater impact on the yield of methane to ethane.
4. There are many formatting issues in the manuscript. The problem of inconsistent brackets and picture number size in Fig.S5-1 and Fig.S6-5
5. There is a subscript problem in the article and supplementary Information, such as line 98.

Response to the reviewers' comments

The authors greatly acknowledge the reviewers for their fruitful comments and suggestions on our manuscript. We revised the manuscript based on their helpful comments. Please find below a detailed response to each comment. The reviewers' comments and the corresponding changes in the revised manuscript are highlighted in blue and red letters, respectively.

Reviewer #1:

General comment: In the manuscript, the authors demonstrate the critical role of interfacial water species in photocatalytic methane conversion in Pt/Ga₂O₃, Pt/NaTaO₃ and Pt/TiO₂ materials through a series of characterization techniques and theoretical calculations. Due to these water-assisted effects, the formation of moderately stable [•]CH₃ radicals in the interface enhances the photocatalytic conversion by more than ~30 times under the conditions of ambient temperature and pressure. However, the quality of the manuscript remains to be improved before its acceptance for publication. Some major comments have been listed for the authors' reference to further improve the quality of this work.

Here are some points should be carefully addressed:

1. In this paper, during the photocatalytic methane conversion reaction, not only the environment of the material is different (dry and wet), but the temperature of its surface is also different (from ~295 K to ~318 K). Does temperature have an effect on catalytic activity? How did the authors rule out thermal effects?

Response: Thank you for reviewing our manuscript and your fruitful comment. First of all, we would like to emphasize that the initial C–H activation of methane (CH₄ → [•]CH₃) is a photocatalytic process driven by photogenerated hole species. Therefore, the slight increase in the sample temperature ($\Delta T \approx 20$ K) cannot induce any methane oxidation reactions under dark conditions.

To demonstrate the above description, we confirmed that no appreciable methane conversion occurs by external heating of the Pt/Ga₂O₃ photocatalyst to ~318 K with a halogen lamp. Note that the wavelength of this halogen lamp (500–3500 nm) is insufficient for exciting valence band electrons of Ga₂O₃ whose band gap energy is corresponding to ~260 nm. As shown in Fig. S9-1, almost no production derived from methane conversion was detected at ~318 K without UV irradiation while H₂ and CO₂ yields evolved linearly under UV irradiation after the induction period* of ~5 min. Therefore, the increase in the sample temperature induced by UV irradiation does not have an influence on the photocatalytic C–H activation of methane.

* An induction period is the initial slow phase of a chemical reaction and is often observed before the reaction system reaches quasi steady-state [*J. Catal.* **338**, 21 (2016)]. Existence of induction periods was often observed on photocatalysis, and the photocatalytic activity was typically evaluated under the quasi steady-state conditions after the induction periods [*J. Phys. Chem. B* **109**, 17212 (2005); *J. Phys. Chem. C* **112**, 5542 (2008); *Top. Catal.* **58**, 70 (2015)].

It is also noteworthy that the reaction temperature rose in the 5–20 min under UV irradiation from ~309 K to ~318 K (Fig. S9-1b) while the CO₂ and H₂ yields linearly increased in this period. This indicates that the photocatalytic activity does not depend on the sample temperature within this temperature range.

Based on the above discussion, we can conclude that there is no thermal assist in the photocatalytic activity. These contents on the thermal effects were added in the supplementary information (Supplementary Section S9).

Revision in the manuscript

S9. Effects of sample heating on photocatalysis.

No appreciable methane conversion occurred by heating of the Pt/Ga₂O₃ photocatalyst to ~318 K with a halogen lamp. Note that the wavelength of this halogen lamp (500–3500 nm) is insufficient for exciting valence band electrons of Ga₂O₃ whose band gap energy is corresponding to ~260 nm. As shown in Fig. S9-1, almost no production derived from methane conversion was detected at ~318 K without UV irradiation while H₂ and CO₂ yields evolved linearly under UV irradiation after the induction period^{21–25} of ~5 min. Therefore, the increase in the sample temperature induced by UV irradiation does not have an influence on the photocatalytic C–H activation of methane.

It is also noteworthy that the reaction temperature rose in the 5–20 min under UV irradiation from ~309 K to ~318 K (Fig. S9-1b) while the CO₂ and H₂ yields linearly increased in this period. This indicates that the photocatalytic activity does not depend on the sample temperature within this temperature range.

Figure S9-1. Time profiles of the sample temperature and the H₂ and CO₂ mass signal produced on Pt/Ga₂O₃ photocatalysts at a methane partial pressure of 70 kPa and water partial pressures of 2 kPa under (a) only halogen lamp irradiation (thermal heating condition; ~318 K) and (b) UV lamp irradiation. Under the UV irradiation, the products were evolved linearly after the induction period of ~5 min, while almost no change on mass signal was detected under the halogen lamp irradiation.

2. Due to these water-assisted effects, the conversion efficiency of methane is improved. At the same time, the three materials do not have high selectivity. How does the author think about this?

Response: Toward sustainable utilization of the most unreactive hydrocarbon of methane, activation of the robust C–H bond under mild conditions is of significance. In this study, we demonstrated that the mediation of interfacial water drastically enhanced the methane conversion rate on non-thermal photocatalysis at ambient conditions (Fig. 1). We also clarified the mechanism of the initial activation ($\text{CH}_4 \rightarrow \cdot\text{CH}_3$) mediated by the water species. On our employed reaction systems, most of the generated $\cdot\text{CH}_3$ was totally oxidized on catalyst surfaces to CO_2 . Meanwhile, CO and ethane formation was also observed as the products of side reactions: partial oxidation and non-oxidative coupling of $\cdot\text{CH}_3$. From our kinetic analysis, stabilization energy of the $\cdot\text{CH}_3$ intermediate (U) for CO_2 , CO, and ethane formation was estimated to be of the order of 40 kJ/mol (Table 1). This indicates the similarity in the molecular-level mechanism of $\cdot\text{CH}_3$ formation as the intermediate of the total oxidation, partial oxidation, and non-oxidative coupling.

Although we focused on the water-assisted methane activation phenomenon and its mechanism in this study, high selectivity of CO or ethane is required to effectively convert the activated $\cdot\text{CH}_3$ to the value-added products. Our experimental results suggest that the preference of the total oxidation, partial oxidation, and non-oxidative coupling of the $\cdot\text{CH}_3$ intermediate species determine the reaction selectivity (Figs. 5a and 5b). Here, we remark that the U values for ethane formation on Pt/Ga₂O₃ and Pt/NaTaO₃ photocatalysts were slightly lower than those for CO_2 and CO formation. This slight difference between the U values would imply that the $\cdot\text{CH}_3$ formation sites are different between ethane formation and CO, CO_2 formation. We would like to elucidate the origin of the difference on the U values in our future work with careful additional experiments. If the slight difference of U is derived from the difference of $\cdot\text{CH}_3$ formation sites, it may be possible to improve the reaction selectivity by increasing the generation sites of $\cdot\text{CH}_3$ radicals that predominantly participate in non-oxidative coupling toward ethane, or by increasing the generation sites of $\cdot\text{CH}_3$ radicals that predominantly participate in oxidation reactions toward CO and CO_2 . This possibility will also be explored in our future work. Thank you for your comment.

3. The authors should state in the article the reasons for choosing Pt/Ga₂O₃, Pt/NaTaO₃ and Pt/TiO₂ for research.

Response: As mentioned in the introduction section of the main text, we employed Ga₂O₃, NaTaO₃, and TiO₂ as representative d¹⁰ (Ga₂O₃) and d⁰ (NaTaO₃ and TiO₂) photocatalysts whose conduction bands are composed predominantly of *sp* and *d* orbitals, respectively [*Chem. Soc. Rev.* **38**, 253 (2009)]. These photocatalyst samples showed stable activities (Fig. S2-3) and significant robustness without catalyst deactivation, e.g., photo-corrosion [*J. Phys. Chem. C* **114**, 11466 (2010); *Chem. Commun.* **56**, 6348 (2020); *Appl. Catal. A-Gen.* **521**, 125 (2016)]. In addition, TiO₂ is a well-known model material in photocatalysis because the TiO₂ photocatalyst has been studied for over half a century [*Chem. Rev.* **119**, 11020 (2019)] since the discovery of photocatalysis [*Nature* **238**, 37 (1972)]. Thus, we chose these three materials as representative photocatalysts for the first demonstration of the effect of interfacial water on photocatalytic methane activation.

To emphasize the above description, we modified the introduction and summary section. We also added the following paragraph in the first section of results and discussion. Thank you for your comment.

Revision in the manuscript

Line 83: Here, we combined real-time mass spectrometry with *operando* infrared (IR) absorption spectroscopy and *ab initio* molecular dynamics (AIMD) simulations to provide microscopic insights into non-thermal photocatalytic conversion of methane. We employed three metal oxides as representative d¹⁰ (Pt/Ga₂O₃) and d⁰ (Pt/NaTaO₃ and Pt/TiO₂) photocatalysts⁸. Systematic experiments on these photocatalysts under controlled pressures of methane gas and water vapor clearly showed that interfacial water species play a key role in the photocatalytic methane conversion under ambient conditions.

Line 97: We employed Ga₂O₃, NaTaO₃, and TiO₂ as representative d¹⁰ (Ga₂O₃) and d⁰ (NaTaO₃ and TiO₂) photocatalysts whose conduction bands are composed predominantly of *sp* and *d* orbitals, respectively⁸. These photocatalyst samples are known to have stable activities and significant robustness without catalyst deactivation, e.g., photo-corrosion^{17,29,30}. In addition, TiO₂ is a well-known model material in photocatalysis because the TiO₂ photocatalyst has been studied for over half a century^{23,31} since the discovery of photocatalysis⁶.

Line 386: In summary, we have demonstrated for the three representative d⁰ and d¹⁰ oxide photocatalysts (Ga₂O₃, NaTaO₃, and TiO₂) with the different band-gap energy that the photocatalytic activation of the robust C–H bond of methane is efficiently accelerated by interfacial water species at ambient temperatures and pressures.

4. This sentence mention “The experimental results of preferential water oxidation on the C-H activation process of methane thus imply the crucial role of interfacial water that is unexplainable under conventional simple assumption based on thermodynamic equilibrium.” But the oxidation potential of water is higher than that of methane in thermodynamics. Please confirm this view.

Response: We would like to apologize for our confusing description in the original manuscript. Once $\cdot\text{OH}$ is generated by photogenerated holes, $\cdot\text{OH}$ has enough oxidizing potential for inducing methane oxidation ($\text{CH}_4 + \cdot\text{OH} \rightarrow \cdot\text{CH}_3 + \text{H}_2\text{O}$), due to the redox potential of the water oxidation (2.73 V vs. SHE) higher than that of methane oxidation (2.06 V vs. SHE).

However, we here intended to discuss which molecule, methane or water, is readily oxidized by the *surface holes* ($\text{h}^+(\text{O}_{\text{lat}})$). Although the holes get more stabilized through methane oxidation ($\text{CH}_4 + \text{h}^+(\text{O}_{\text{lat}}) \rightarrow \cdot\text{CH}_3 + \text{H}^+$, 2.06 V vs. SHE) than water oxidation ($\text{H}_2\text{O} + \text{h}^+(\text{O}_{\text{lat}}) \rightarrow \cdot\text{OH} + \text{H}^+$, 2.73 V vs. SHE) from the thermodynamic point of view, our experimental results (Figs. 1 and 2) showed that the oxidation of water precedes that of methane. This indicates that the photocatalytic oxidation by the surface holes is under kinetic control, rather than thermodynamic control. This feature is also supported by our MD simulations which revealed that the barrier for water activation is lower than that for methane activation under the wet condition (Fig. 3a in the main text and Fig. S4-2a in the supplementary information) and that water has higher accessibility to the surface holes than methane.

Based on the reviewer’s comment, we modified the main text to clarify our intention as follows. Thank you for your comment.

Revision in the manuscript

Line 206: Notably, the kinetic isotope effects on methane conversion and total hydrogen production were negligible (Fig. S2-2). This result indicates that water activation (equation (2)) does not determine the reaction rate in the methane photocatalytic reactions, which is contrary to water splitting, where water activation has been considered to be a rate-determining step³⁹. Since the redox potential of water oxidation ($E^\circ_{\cdot\text{OH}/\text{H}_2\text{O}} = 2.73$ V vs. the standard hydrogen electrode (SHE))⁴⁰ is higher than that of methane oxidation ($E^\circ_{\cdot\text{CH}_3/\text{CH}_4} = 2.06$ V vs. SHE)⁴¹, it is simply assumed from the thermodynamic point of view that the photogenerated holes get more stabilized by the oxidation of methane than water. In contrast to the thermodynamic tendency, however, our experimental results (Figs. 1 and 2) indicate that the holes at the oxide surfaces preferentially oxidize water rather than methane, and then the preactivated water species activate the C–H bond of methane. This implies that photocatalytic preferential oxidation of water over methane by the surface holes is induced kinetically rather than thermodynamically.

5. The authors only studied the role of interfacial water in three materials Pt/Ga₂O₃, Pt/NaTaO₃ and Pt/TiO₂. Then the interfacial water only has an effect on the three materials or is it universal.

Response: We chose the three typical oxides with different physical properties, such as band gap energy and the number of d-electrons. The sample-independent results indicate the common effects of the interfacial water at least among the three employed materials.

Based on the mechanism of the water-assisted methane activation on the three photocatalysts, we can rationally assume that the effect of the interfacial water on photocatalytic methane activation is a common phenomenon for most of the metal oxide photocatalysts, as follows. Since valence bands of metal oxides mainly consist of O2p orbitals, the valence band maximums of metal oxides exist at a similar depth (~ 3 eV from the SHE). Furthermore, most of the metal oxide surfaces interact with methane molecules quite weakly in comparison to water molecules: thus, methane molecules cannot substantially adsorb on surfaces without UV irradiation. Therefore, the interfacial water molecules have much opportunity to accept photogenerated holes from the surfaces in comparison with gaseous methane molecules, and are kinetically advantageous for the hole-driven oxidation. We thus consider that the water-assisted effects would be universal among the metal oxide photocatalysts with these features.

Based on the reviewer's comment, we added the following paragraph ahead of the summary paragraph.

Revision in the manuscript

Line 375: We remark finally that the effect of the interfacial water on photocatalytic methane activation would be a common phenomenon for most of the metal oxide photocatalysts. Since valence bands of metal oxides mainly consist of O2p orbitals, the valence band maximums of metal oxides exist at a similar depth (~ 3 eV from the SHE). Furthermore, most of the metal oxide surfaces interact with methane molecules quite weakly in comparison to water molecules: thus, methane molecules cannot substantially adsorb on surfaces without UV irradiation. Therefore, the interfacial water molecules have much opportunity to accept photogenerated holes from the surfaces in comparison with gaseous methane molecules, and are kinetically advantageous for the hole-driven oxidation. The water-assisted effects would be universal among the metal oxide photocatalysts with these features.

Reviewer #2:

General comment: In the manuscript entitled "Critical impacts of interfacial water on the photocatalytic C-H conversion of methane, the authors use IR spectroscopy combined with mass spectrometry supported by simulations to address the role of interfacial water on the conversion of methane over Pt supported on three different support materials.

The manuscript is well-written and authors present a rigorous piece of work and get the maximum out of particularly the IR results. They explain the data in great detail and I cannot discredit these results in any way.

What I really miss in the manuscript is structural information. No structural information is presented at all (XRD, SEM, TEM). Such data would make the story easier to follow and could justify the assumptions in the manuscript. At the very least, include a sketch in the beginning of the manuscript to allow the reader to understand the structural model of the catalytic systems in question.

Recommendation: publish after minor revision.

Response: Thank you for reviewing our manuscript and we acknowledge the reviewer for the positive evaluation of our work. Based on your fruitful suggestions, we added the SEM and TEM images of the three samples in the supplementary information (Supplementary Section S8).

Revision in the manuscript

S8. SEM and TEM images of the photocatalyst samples.

Figure S8-1. Scanning electron microscopy (SEM) images of (a) Ga₂O₃, (b) NaTaO₃ photocatalysts, and a tunneling electron microscopy (TEM) image of (c) TiO₂ (ST-01) photocatalysts. The typical particle size is ~3 μm (Ga₂O₃), ~200 nm (NaTaO₃), and ~3 nm (TiO₂), respectively.

Reviewer #3:

General comment: The authors combined real-time mass spectrometry with operando infrared (IR) absorption spectroscopy and ab initio molecular dynamics (AIMD) simulations to provide microscopic insights into non-thermal photocatalytic conversion of methane. Systematic experiments on representative photocatalysts (d^{10} photocatalyst: Pt/Ga₂O₃, d^0 photocatalysts: Pt/NaTaO₃ and Pt/TiO₂) under controlled pressures of methane gas and water vapor clearly showed that interfacial water species play a key role in the photocatalytic methane conversion under ambient conditions, but there are some issues needed to be further emphasized before it is suitable for the publication. A major revision is needed, the details as follows:

1. The authors carried out systematic experiments on representative photocatalysts (d^{10} photocatalyst: Pt/Ga₂O₃, d^0 photocatalysts: Pt/NaTaO₃ and Pt/TiO₂), the data of photocatalyst: Pt/TiO₂ should be provided in figure.

Response: Thank you for reviewing our manuscript. Based on the reviewer's comment, we added the data of *operando* IR spectroscopy for Pt/TiO₂ samples in Fig. 2 in the revised manuscript.

Revision in the manuscript

Figure 2 | Operando infrared (IR) spectroscopy of photocatalytic methane activation with isotope-labeled water (D₂O). (a–c) Time evolution of the IR spectra in the O–H stretching region for the (a) Pt/Ga₂O₃, (b) Pt/NaTaO₃, and (c) Pt/TiO₂ photocatalysts under ultraviolet irradiation at a CH₄ pressure of 30 kPa under wet conditions (2 kPa of D₂O). The *operando* IR measurements were started from the time when the temperature increase leveled off in order to extract the response derived from photocatalytic reaction while excluding the spectral change derived from the sample heating (see Supplementary Section S9 for details). Time zero ($t = 0$ min) was defined as the starting time for the IR measurement. (d–f) Time evolution of the peak height at 3250 cm⁻¹ for the (d) Pt/Ga₂O₃, (e) Pt/NaTaO₃, and (f) Pt/TiO₂ photocatalysts at CH₄ pressures of 5, 30, 70 kPa. (g–i) Growth rate of the peak at 3250 cm⁻¹ (left axis) and CH₄ total conversion rate (right axis) on the (g) Pt/Ga₂O₃, (h) Pt/NaTaO₃, and (i) Pt/TiO₂ photocatalysts.

2. The author should explain the “15 min line” lacking in Fig. 2(c) and the difference between $P_{\text{CH}_4}=6$ in Fig. 2(b) and $P_{\text{CH}_4}=5$ in Fig. 2(d) in this article.

Response: First of all, we would like to apologize for a typo of the P_{CH_4} value. The time profiles of IR peak shown in Fig. 2 were measured at $P_{\text{CH}_4}=5, 30,$ and 70 kPa for all the photocatalyst samples. We revised the P_{CH_4} value in Fig. 2d from 6 kPa to 5 kPa. Thank you for pointing it out.

Concerning the lack of “15 min line”, we would like to mention that the sample temperature was increased by UV irradiation and the temperature increase leveled off at ~ 15 min (please see Fig. S9-1b in the supplementary information or the response to the reviewer #1). As described in our previous study [*Vac. Surf. Sci.* **63**, 476 (2020)], absorbance change signals derived from thermally excited electrons and thermally desorbed water molecules are also observed on the *operando* IR spectrum measured during the increase in the photocatalyst sample temperature. Therefore, we started the *operando* IR measurements from the time when the temperature increase leveled off in order to carefully extract the response derived from photocatalytic reaction while excluding the spectral change derived from the sample heating. Notably, one layer of adsorbed water molecules covers the sample at the reached temperature of ~ 318 K as described in Supplementary Section S1, and the photocatalytic activity was evaluated at this period in this study.

Because the time needed for the sample temperature to level off slightly depends on the photocatalyst samples, the starting time for the IR measurement should also be adequately changed depending on the samples. To simplify the description and avoid confusion among the readers, we displayed the *operando* IR spectra (Fig. 2 in the revised manuscript) defining $t = 0$ as the starting time for the measurement.

These contents were added in the revised caption of Fig. 2 and the supplementary information (Supplementary Section S9). Thank you for your comment.

Revision in the manuscript

<Caption of Fig. 2>

Figure 2 | *Operando* infrared (IR) spectroscopy of photocatalytic methane activation with isotope-labeled water (D_2O). (a–c) Time evolution of the IR spectra in the O–H stretching region for the (a) Pt/Ga₂O₃, (b) Pt/NaTaO₃, and (c) Pt/TiO₂ photocatalysts under ultraviolet irradiation at a CH₄ pressure of 30 kPa under wet conditions (2 kPa of D₂O). **The *operando* IR measurements were started from the time when the temperature increase leveled off in order to extract the response derived from photocatalytic reaction while excluding the spectral change derived from the sample heating (see Supplementary Section S9 for details). Time zero ($t = 0$ min) was defined as the starting time for the IR measurement.** (d–f) Time evolution of the peak height at 3250 cm⁻¹ for the (d) Pt/Ga₂O₃, (e) Pt/NaTaO₃, and (f) Pt/TiO₂ photocatalysts at CH₄ pressures of 5, 30, 70 kPa. (g–i) Growth rate of the peak at 3250 cm⁻¹ (left axis) and CH₄ total conversion rate (right axis) on the (g) Pt/Ga₂O₃, (h) Pt/NaTaO₃, and (i) Pt/TiO₂ photocatalysts.

S9. Effects of sample heating on photocatalysis.

The sample heating derived from UV irradiation (Fig. S9-1b) also affects the *operando* IR spectra (Fig. 2). As described in our previous study²⁶, absorbance change signals derived from thermally excited electrons and thermally desorbed water molecules are also observed on the *operando* IR spectrum measured during the increase in the photocatalyst sample temperature. Therefore, the *operando* IR measurements were started from the time when the temperature increase leveled off in order to carefully extract the response derived from photocatalytic reaction while excluding the spectral change derived from the sample heating. Notably, one layer of adsorbed water molecules covers the sample at the reached temperature of ~318 K as described in Supplementary Section S1, and the photocatalytic activity was evaluated at this period in this study.

Because the time needed for the sample temperature to level off slightly depends on the photocatalyst samples, the starting time for the IR measurement should also be adequately changed depending on the samples. Thus, time zero ($t = 0$ min) was defined as the starting time for the measurement.

3. The authors think the kinetic isotope effects on methane conversion and total hydrogen production were negligible, the authors should explain why the use of water with and without isotopes in Figure S2-2(b) has a greater impact on the yield of methane to ethane.

Response: First of all, we would like to emphasize that the ethane formation rates themselves account for at most 5% of the total methane conversion rates and kinetic isotope effects are almost negligible on the total methane conversion rates and the hydrogen formation rates. Therefore, we neglected the kinetic isotope effects on the ethane formation rates.

As the reviewer pointed out, it is possible that the substantial isotope effects of a few tens of percent exist on the ethane formation. Because this isotope effect of ethane is minor in the total methane conversion and does not affect our main conclusions and concepts, we would like to discuss this issue in our future work with careful additional experiments. Thank you for your comment.

4. There are many formatting issues in the manuscript. The problem of inconsistent brackets and picture number size in Fig.S5-1 and Fig.S6-5.
5. There is a subscript problem in the article and supplementary Information, such as line 98.

Response: We corrected all of these formatting problems in the main text and the supplementary information. Thank you for your comments.

REVIEWERS' COMMENTS:

Reviewer #1 (Remarks to the Author):

Thanks to the author for point-by-point response to the comments. There are still many errors in the article. The pictures in the authors' manuscripts are very ugly. Please carefully check the clarity, aspect ratio and borders of each picture. It is recommended to accept after solving this problem.

Reviewer #2 (Remarks to the Author):

The authors have done a heroic effort in updating the manuscript according to the referee reports and explained their efforts adequately. However, the structural information presented in section S8 is barely mentioned in a single line. Use the data actively in the manuscript. Also, the TEM image in Section S8 seems cropped out from a different manuscript. The quality is awful and the 3 nm particles mentioned in the caption cannot be seen at all. I think this should be updated prior to publication.

Reviewer #3 (Remarks to the Author):

The authors emphasized the reviewers' comments well, it is suitable for publication in the journal

Response to the reviewers' comments

Thank you for reviewing our revised manuscript. We further revised the manuscript based on their comments. Please find below a detailed response to each comment.

Reviewer #1:

Thanks to the author for point-by-point response to the comments. There are still many errors in the article. The pictures in the authors' manuscripts are very ugly. Please carefully check the clarity, aspect ratio and borders of each picture. It is recommended to accept after solving this problem.

Response: We acknowledge the reviewer for reviewing our manuscript and apologize for the unsharp figures. We carefully checked and revised the figures. The submitted figure files get clear and meet the publication criteria. Thank you for your comments.

Reviewer #2:

The authors have done a heroic effort in updating the manuscript according to the referee reports and explained their efforts adequately. However, the structural information presented in section S8 is barely mentioned in a single line. Use the data actively in the manuscript. Also, the TEM image in Section S8 seems cropped out from a different manuscript. The quality is awful and the 3 nm particles mentioned in the caption cannot be seen at all. I think this should be updated prior to publication.

Response: We acknowledge the reviewer for reviewing our manuscript and apologize for the blurred TEM image. We changed the TEM image of TiO₂ to the sharp one as below. In addition, we added the sentence: "The SEM and TEM images of these photocatalyst samples are shown in Figure S4-1." in the methods section. Thank you for your comments.

Revision in the manuscript

Figure S4-1

Reviewer #3:

The authors emphasized the reviewers' comments well, it is suitable for publication in the journal.

Response: Thank you for reviewing our manuscript and the positive reply.